# Computational Approach to Drug Penetration across the Blood-Brain and Blood-Milk Barrier Using Chromatographic Descriptors

**DOI:** 10.3390/cells12030421

**Published:** 2023-01-27

**Authors:** Wanat Karolina, Rojek Agata, Brzezińska Elżbieta

**Affiliations:** Department of Analytical Chemistry, Faculty of Pharmacy, Medical University of Lodz, 90-419 Lodz, Poland

**Keywords:** blood-brain barrier, breast milk barrier, milk-to-plasma ratio, drug pharmacokinetics, protein binding, analytical model

## Abstract

Drug penetration through biological barriers is an important aspect of pharmacokinetics. Although the structure of the blood-brain and blood-milk barriers is different, a connection can be found in the literature between drugs entering the central nervous system (CNS) and breast milk. This study was created to reveal such a relationship with the use of statistical modelling. The basic physicochemical properties of 37 active pharmaceutical compounds (APIs) and their chromatographic retention data (TLC and HPLC) were incorporated into calculations as molecular descriptors (MDs). Chromatography was performed in a thin layer format (TLC), where the plates were impregnated with bovine serum albumin to mimic plasma protein binding. Two columns were used in high performance liquid chromatography (HPLC): one with immobilized human serum albumin (HSA), and the other containing an immobilized artificial membrane (IAM). Statistical methods including multiple linear regression (MLR), cluster analysis (CA) and random forest regression (RF) were performed with satisfactory results: the MLR model explains 83% of the independent variable variability related to CNS bioavailability; while the RF model explains up to 87%. In both cases, the parameter related to breast milk penetration was included in the created models. A significant share of reversed-phase TLC retention values was also noticed in the RF model.

## 1. Introduction

The bioavailability of the blood-brain barrier (BBB) is denoted by the BBB+/− index [1,2]. The classification of BBB+ and BBB− types can be based on calculated log BB values and their comparison with in vivo or in vitro biological test data. Log BB is the most commonly used measure of drug penetration into the central nervous system (CNS), and it is defined as the logarithm of the ratio of drug concentration in the brain to drug concentration in blood [3,4]. Biological experiments determining the level of blood-brain distribution are the best source of information; however, these are extremely time-consuming, costly, and difficult-to-access experiments that require extensive screening.

Another classification which can be found in the literature is CNS+/− [5]. The drug types CNS+ and CNS− do not correspond to the BBB+/− index. CNS inactive drugs include both those that cross the BBB (without appropriate biological targets) and those that do not penetrate the barriers [6,7]. It is more certain to identify BBB+ drugs among CNS+ compounds than to identify BBB- among CNS− [6,8]. There are also studies [9] for which the basis for BBB+ classification was a higher concentration of the drug in the brain than in the blood.

The excretion into breast milk can be described by the M/P parameter, which is the ratio of the drug concentration in milk and plasma. Determining this parameter in vivo is even more difficult than log BB, while its knowledge may increase the safety of pharmacotherapy in breastfeeding women. Difficulties in obtaining the M/P ratio in vivo are mainly due to ethical reasons and the danger of affecting the breastfed infant, therefore most approaches to the subject take the form of case studies. Only a few results are obtained for individual drugs and these may not reflect what is happening in the scale of the entire population. The problem is even greater with drugs newly introduced to the market [10]. The computational methods could be helpful in the estimation of M/P ratios based on API properties and general principles of permeation through biological membranes.

Blood-brain and blood milk-barrier have many differences; the blood-milk barrier is simpler, the main obstacle is the lactocyte membrane. Blood-brain barrier exhibits an additional enzymatic activity and a reinforced structure (Table 1) [11]. CNS-active drugs have been the subject of case studies for safety in the breastfed child; most of them willingly pass into breast milk, and some even have M/P ratios above 1 (e.g., citalopram, escitalopram or sertraline), which indicates a higher concentration in milk than in plasma [12,13].

A small group of active pharmaceutical ingredients APIs (37 compounds) with known biological effects were analysed. The main aim of the study is to investigate the use of chromatographic systems describing the protein binding capacity as analytical models for two phenomena—drug distribution to the CNS and breast milk, and thus to find a relationship between the bioavailability of APIs in the CNS, and the dangers of their use in breastfeeding women. The chemometric analyses were based on the result of an earlier published work [14] in which it was proposed to investigate the aforementioned relationship; however, compounds **1**–**37** were never subject to a separate computational experiment. The structures of 37 APIs, along with their applied physicochemical and biological properties, are included in the Appendix A.

Plasma protein binding (PPB) is one of the most important pharmacokinetic aspects limiting permeation across membranes and biological barriers. Human serum albumin is the most abundant plasma protein and is responsible for the greatest extent for PPB. Therefore, chromatography-based laboratory models have been developed to estimate the level of HSA binding and its effect on CNS entry. An observation was made of the effect of stationary phase modification with bovine serum albumin (BSA) in thin layer chromatography (TLC) and human serum albumin (HSA) in high performance liquid chromatography (HPLC). BSA is considered to be a cheaper alternative to HSA, with similar binding and physicochemical properties [15].

Due to the different routes of entry of drugs into the brain and milk, an immobilized artificial membrane (IAM) column was also used. IAM chromatography is based on the permeability of a compound through a biological membrane; the phospholipid monolayer is bound to a silica base [16].

Two sets of thin layer affinity chromatography were applied: in normal (NP) and reverse phase (RP), and two mentioned columns in the HPLC technique.

## 2. Materials and Methods

The results of chromatographic experiments, as well as molecular descriptors (MDs) connected to the physicochemical properties of APIs were introduced into the research (Table 2). Physicochemical properties were calculated (Hyperchem; HyperCube Inc., Waterloo, ON, Canada, 2002 and ACD Labs; Advanced Chemistry Development Inc., Toronto, ON, Canada) or collected from online databases: Drugbank [17] and Chembl [18]. Chromatographic data: R_f_ and log k parameters were obtained experimentally; R_f_ (retardation factor) is the ratio of the mobile phase distance travelled to the API distance travelled on the TLC plate. HPLC parameter: log k (logarithm of the retention factor), where k is the ratio between the amount of API in the stationary phase to its amount in the mobile phase, and it was obtained from the equation k = (t_R_ − t_M_)/t_M_ (t_R_—retention time; t_M_—void time). TLC and HPLC experiments are detailed in Appendix C.

## 3. Results

### 3.1. Correlation Analysis and Multiple Linear Regression

The limit values of the log BB index are different for many of the proposed BBB penetration prediction models [8,9,24]. The log BB = −0.52 is the logical division into BBB+ and BBB–. The logarithmic value of −0.52 corresponds to a 30% brain–to–plasma ratio of the compound [25]. The optimal classification threshold is usually 0 to −1 [25,26]. Studies conducted earlier defined the BBB+ limit as log BB ≥ −0.9 [27]. The calculations in this study use all the rates of BBB penetration through the 37 observed drugs. The parameter used (B2) is based on the log BB and describes the division into BBB +/− [19], and the following limit values were also used: B2 > −0.90 and B2 > −0.52. Another applied descriptor: CNS +/− is described in the literature [17].

In order to combine two pharmacokinetic phenomena—the drug penetration of blood–milk and the blood–brain barrier, the observation of molecular descriptors of 37 compounds was introduced. MDs which, according to the literature, usually affect BBB +/− and M/P are: binding to plasma proteins—PB, the acid–base nature of the compounds—acid/base and PhCharge—physiological charge. The correlation matrix that describes the relationships between these indicators in the study group is presented in Table 3.

The connection of B2, B2 > −0.90 descriptors and PB is clearly visible. Additionally, it was assumed that the best correlation of the binding strength with the plasma proteins is connected with the B2 descriptor. On its basis, the availability of drugs to the brain can be determined. The choice between B2 > −0.90 and B2 > −0.52 is subjective, so the range of drug properties related to the B2 measure was investigated.

In milk penetration correlation analyses (Table 4), APIs indicated as CNS− were removed from a group of 37 cases. Only potentially CNS+ remain, in line with the theory that centrally acting drugs are the most dangerous. There are 25 such CNS+ compounds, but the introduced LLL H, LactMed, and PB indicators further reduced this number (n = 21).

The significant correlation coefficient (inverse relationship) between the bioavailability of the drug in breast milk M/P with the value of PB_code_ and PB (R = 0.52 and R = −0.42 respectively) indicates that the high level of protein binding reduces the risk of drug penetration into breast milk. The danger of treatment during the lactation period is associated with the nature of biologically active compounds: acids (a; code: −1) are the safest. They strongly bind to human plasma albumin. The bases (b; code: 1) are the least secure. Here, the ratio of milk content (M) to plasma content (P) will be the greatest. The correlation matrix also shows that the relationship between the acid/base, B2 and M/P is very similar (R = 0.54). Such a result confirms the relationship of drug binding power with proteins and their bioavailability in compartments isolated by the blood–brain and blood–milk barriers.

The main task was to study the interrelationships and relationships between the bioavailability of drugs in the CNS and the dangers of their use in women during breastfeeding. Therefore, the MLR analysis was performed for the dependent variable B2. A summary of the regression obtained is presented below. The test result is very good, not only because of the high correlation coefficient R = 0.91, which explains 83% of the total variability of B2 (n = 33), but also due to the introduction of the M/P parameter to the model. The relationship between B2 and M/P is, as expected, directly proportional (Figure 1, Equation (1)).
B2 = −0.33(±0.19) −0.25(±0.02) HA + 0.15(±0.04) Sa + 0.17(±0.08)M/P(1)

R = 0.91, R^2^ = 0.83, F(3,29) = 47.05, *p* < 0.00000, s = 0.2811; n = 33Q^2^_LOO_ = 0.78, SDEP = 0.2839, PRESS = 2.893, S_PRESS_ = 0.2875, Q^2^_LMO_ = 0.80

The B2 index is most strongly represented by the number of hydrogen bond acceptors (HA) and the molecular surface area of the drug (Sa). Interesting results were also obtained from the MLR model of the dependent variable PB (Equation (2)).
PB = 0.91(±0.41) + 0.014(±0.031)logD − 0.15(±0.066)M/P + 0.14(±0.050)Sa − 0.073(±0.027) HA + 0.058(±0.040)eH-eL(2)
R = 0.76, R^2^ = 0.57, F(5,27) = 7.244, *p* < 0.00020, s = 0.2417

The repetition of all MDs associated with the B2 value (HA, M/P, Sa) in the resulting PB model suggests the inclusion of the chromatographic data connected with protein binding in explaining the penetration of APIs into the CNS and breast milk. Due to this connection, the chromatographic systems describing the protein binding capacity became analytical models for two pharmacokinetic phenomena—the distribution to the CNS and milk.

Further analyses were based on the chromatographic data: R_f_ values from TLC (NP and RP mode) and HPLC retention times were collected using two columns: HSA and IAM. Initially, a study of the results of using chromatographic data for a set of CNS +/− cases (n = 37) was performed and presented in the following correlation matrices (Table 4), then all types of chromatographic systems used were checked for the group of bioavailable CNS+ (n = 25) (Table 5).

Both groups of cases (CNS+/− and CNS+ only) clearly associate bioavailability in breast milk with the behavior of compounds tested in a thin layer chromatography environment. They apply both to the stationary phase, obtained by modifying silica gel plates (NP), and to silica gel plates, which are silanized (RP). Slight differences between the groups of cases—CNS+/−, CNS+ result from the size of the groups. The results of experiments using HPLC yielded worse results.

An additional modifier was the R_f_ value of drugs on non–protein coated plates (control plate—C). The introduction of the R_f_/C parameter—NP/C and RP/C—allows the observation of chromatographic effects related only to the presence of the protein in the stationary phase. The modified data are presented separately in the following correlation matrices from various chromatographic experiments (Table 6).

The observation of the correlation of chromatographic data by type of chromatography indicates RP TLC as the system that best describes the characteristics of drug excretion into breast milk in the group of CNS-active APIs. The RP/C descriptor also has the highest correlation with M/P (R = −0.57).

### 3.2. Random Forest Regression

RF regression analyses were performed using seven MDs. Four models were created, each time using different chromatographic data (Rf values from NP TLC and RP TLC or log k obtained from HPLC-HSA, HPLC-IAM experiments). The dependent variable was the B2 parameter, similar to the MLR analysis. The molecular descriptors that gave the best results and therefore were used in the models are listed in the Table 7:

The predictor importance plot revealed that TLC data (MD no. 7 on the plots), with an indication of the reversed phase system, had the greatest contribution to the creation of the RF model. This impact is comparable to the contribution of log P or HD. HPLC-HSA log k values show a significantly lesser effect on regression, and HPLC-IAM data indicates a negligible influence. (Figure 2).

The most important MD in RF models is the number of hydrogen bond acceptors (HA). Only PB reveals an inversely proportional relationship, which is the expected conclusion. The M/P parameter shows a similar contribution as physicochemical MDs such as PhCharge or log P. This information creates another link between crossing the BBB and the blood-milk barrier.

RF regression models were ultimately built with only TLC data, as reported by the predictor importance plots. R^2^ and Q^2^ values oscillated around 0.85–0.87 and 0.78–0.80, respectively (Figure 3 and Figure 4), which is a satisfactory result. The best fit of the model can be seen in the B2 value range from −1.5 to 0; extreme values are less correlated.

With each model calculation in the RF method, different decision trees are involved. Therefore, the RF regression was repeated 20 times to obtain averaged results. Table 8 shows the mean values of R^2^ and Q^2^ from the models. The data from NP TLC gives slightly higher results of the model and cross-validation determination coefficients, but it is not a significant difference.

### 3.3. Cluster Analysis

This analysis was performed to compare the variability of the blood–brain barrier permeation parameters and the chromatographic data describing them.

The parameter CNS+/− was used as an indicator of the bioavailability of drugs in the brain. As an indicator of pharmacotherapy safety in the feeding period, the M/P code parameter was used, which was created by scaling the M/P parameter. The M/P_code_ parameter ranges from 1–4. The values 1, 2, 3 and 4 have cases where the determined M/P level does not exceed 0.3, 0.8, 1.1 and 4, respectively. The logical conclusion also remains that drugs with a lower concentration in milk than in blood can be considered safer (M/P−). If the M/P value exceeds 1, the use of pharmacotherapy in a nursing woman is inadvisable or prohibited (M/P+). The study group was observed with the possibility of dividing it into two or four clusters: drugs CNS+ and CNS− and M/P_code_ 1–4, according to the level of safety of use.

The application of two clusters (Appendix B Figure A1) is not effective. The CNS+/− takes the value of 0.5 and 0.8 for clusters with values 0 and 1, and all M/P values, in both clusters that are below 1 (instead of <1 and >1); this means that both clusters gather mixed cases. Such analysis is not effective. This is due to the considerable complexity of both phenomena, which are not obviously defined, even though biomedical models.

The splitting of the cases into four clusters improved the result (Figure 5). In the case of M/P values, the four clusters split into two groups—clusters 2 and 4 have an M/P− < 1, and clusters 1 and 3 are M/P+ > 1. At the same time, clusters 1 and 3 showed M/P+ (3.4–3.8) values, and clusters 2 and 4 corresponded to M/P− (1.1–2.2). The definition of the CNS+/− value strictly follows this division. An interesting observation is the distribution of the values of the chromatographic data. In both TLC and HPLC formats, all extreme data (the smallest and largest R_f_ and log k values) correspond to clusters 1 and 3. The intermediate clusters are 2 and 4.

Assuming that the inaccuracy of the division of cases into M/P+ and M/P− may be caused by a particularly difficult and ambiguous differentiation of CNS+/− APIs, a grouping analysis of k–means excluding CNS− drugs was performed.

An M/P value of less than 1.0 shows that only minimal amounts of drugs are transferred into the milk; these types of drugs are classified as low risk (LR). Drugs with an M/P value of 1.0 or more may be present in breast milk at a higher concentration than in the mother’s plasma and are classified as high risk (HR). Thus, this leads to more drug being transported into the infant’s body and, consequently, to side effects [12].

As the results of these experiments depend on various uncontrolled variables such as laboratory conditions, geographic region and lactation time, the reliability of these experimental data is questionable. In addition, there are many drugs for which M/P ratios have not been determined, so it is necessary and useful to develop some theoretical methods such as the Quantitative Structure Property Relationship (QSPR) for predicting M/P values. In QSPR approaches, the chemical structures of compounds are quantitatively correlated with their biological activity [28].

In the current analysis, an additional parameter, HR/LR, was introduced, where HR—high risk (code value 1) of passing the drug into milk. The second value is LR–low risk (code with a value of 0). The result of the analysis, with the application of the 4 CNS+ APIs clusters, are shown in the graph below (Figure 6).

## 4. Discussion

The inversely proportional relationship between the strength of API and protein binding and its potential presence in the CNS has been demonstrated in the presented models. There is also a strong similarity between the acid/base and PB descriptors, and M/P milk penetration (correlation 0.54 and −0.42) or B2 brain penetration (correlation 0.54 and −0.31). This confirms the relationship between API-protein binding and their bioavailability in compartments isolated by blood-brain and blood-milk barriers.

The study reveals that the chromatographic systems can be analytical models for two different pharmacokinetic phenomena—distribution into the CNS and breast milk. The idea of analytical models for imaging pharmacokinetic properties can be based in both cases on the observation of the affinity of APIs to plasma proteins. An analysis of the chromatographic data with the division into the types of chromatography indicates TLC as the system that better describes the characteristics of drug penetration into breast milk when compared with liquid chromatography. The RP TLC R_f_ value was the most useful in an RF regression model with B2 as the BBB penetration parameter; RP/C has the highest correlation (−0.57) with the M/P descriptor.

M/P was involved in both the MLR and RF regression models of BBB penetration, and the relationship was directly proportional. This may lead to the assumption that the use of CNS+ drugs in breastfeeding women is more dangerous. However, due to difficulties in experimentally determining the M/P values, there is uncertainty in the reported M/P values which may influence the results of QSPR modelling. Therefore, classifying drugs into the high-risk (HR) and low-risk (LR) classes is more appropriate than accurately predicting their M/P values.

## Figures and Tables

**Figure 1 cells-12-00421-f001:**
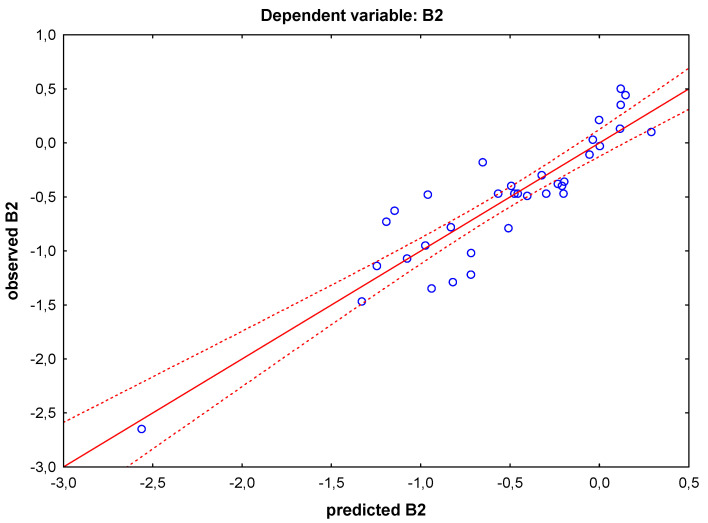
Scatter plot of the relationships between the predicted and observed variables B2 in the MLR model.

**Figure 2 cells-12-00421-f002:**
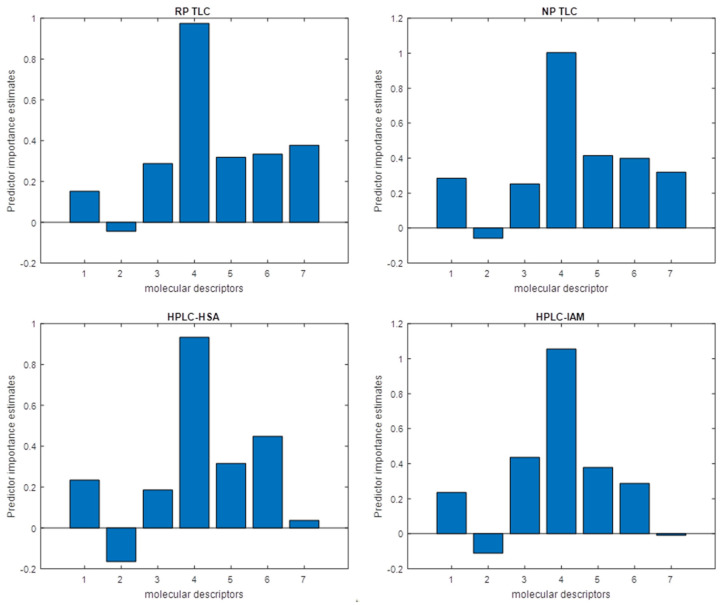
Model predictor importance plots. MDs are listed in accordance with Table 5. MD no. 7 is the value obtained in chromatographic experiments: R_f_ or log k.

**Figure 3 cells-12-00421-f003:**
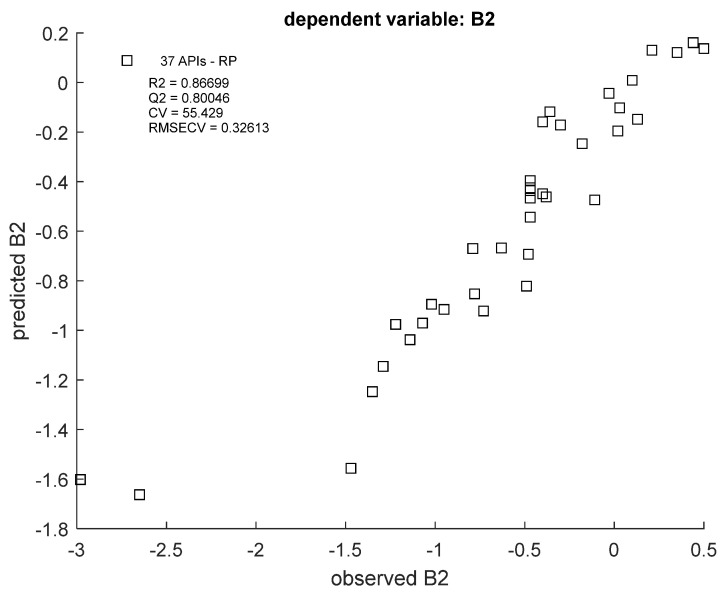
Scatter plot of predicted vs. observed B2 values in RF model, using RP TLC data. R2 = coefficient of determination; Q2 = coefficient of determination for the cross-validated models; RMSECV = root-mean-square error of cross-validation.

**Figure 4 cells-12-00421-f004:**
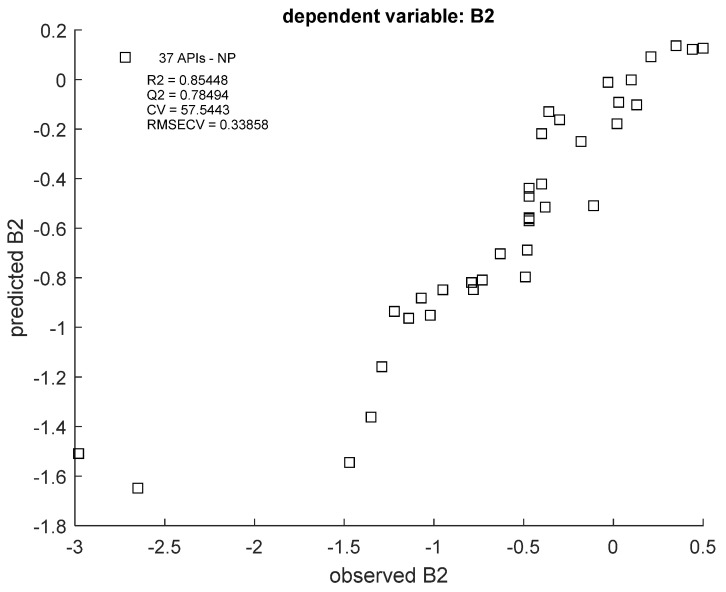
Scatter plot of predicted vs. observed B2 values in an RF model using NP TLC data. R2 = coefficient of determination; Q2 = coefficient of determination for the cross-validated models; RMSECV = root-mean-square error of cross-validation.

**Figure 5 cells-12-00421-f005:**
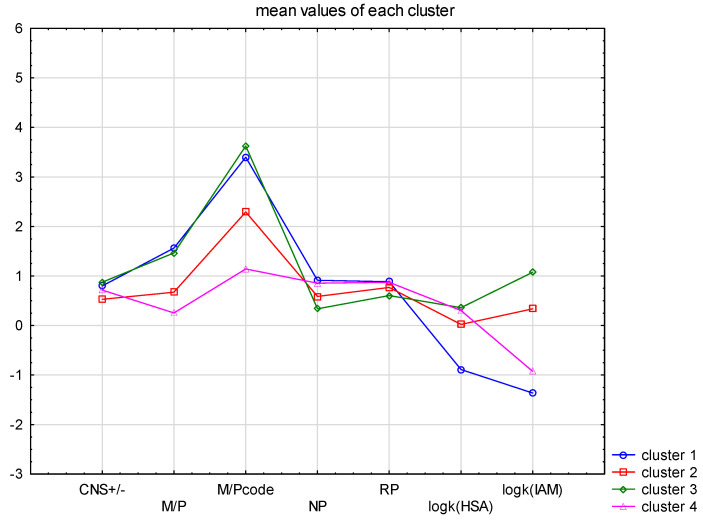
Mean value graph of the chromatographic data for four clusters: CNS+, CNS−, M/P+ and M/P−.

**Figure 6 cells-12-00421-f006:**
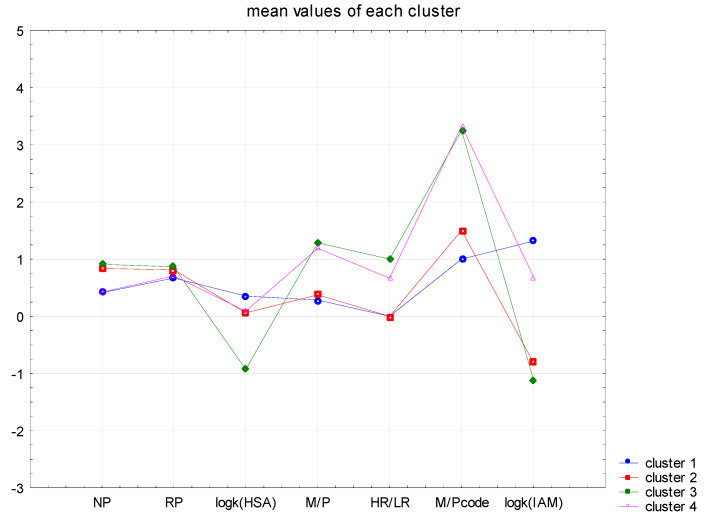
Graph of the mean values of the chromatographic data for four clusters of CNS+ drugs.

**Table 1 cells-12-00421-t001:** Differences and similarities between blood-brain barrier and blood milk barrier.

	Blood-Brain Barrier	Blood-Milk Barrier
**structure**	double phospholipid membrane with additional tight (TJs) and adherence junctions (AJs) in the basement membrane	double phospholipid membrane the first days of lactation: large cellular spaces are formed between lactocytes, which can be easily penetrated by compounds from blood vessels and lymph
**function**	protection of the CNS against the penetration of unnecessary and toxic substances and selective supply of necessary ones (glucose, nutrients, oxygen)	passing nutrients into milk, and retaining unnecessary substances (mainly macromolecular ones)
**additional activity**	P-glycoprotein activity specific enzymatic activity—breakdown of chemical compounds system of active removal of substances (brain-to-blood efflux), carried out by transporters	not observed
**mechanisms of penetration**	simple diffusion membrane transporters endocytosis (often via receptors)	mainly simple diffusion secretory route (beginning of the lactation, depends on the structure of the mammary gland)

**Table 2 cells-12-00421-t002:** List of molecular descriptors (MDs) used in statistical analyses.

MD	Description	Source
acid/base	acidic, basic or neutral nature of an API	Chembl
B2	computational parameter, determines penetration through the blood-brain barrier: log bb = 0.547–0.016 PSA	[19]
CNS+/−	describes the bioavailability in the CNS	Drugbank
eL-eH	ionisation capacity	Hyperchem
HA	hydrogen bonds acceptors	Hyperchem
HD	hydrogen bonds donors	Hyperchem
LactMed	describes the bioavailability in the breastmilk (code: 0–2)	[20]
LLL H	Hale’s toxicity scale (code: 1–5)	[21]
log D	distribution coefficient	ACD Labs
log k_HSA_	logarithm of retention factor from HPLC_HSA_	HPLC
log k_IAM_	logarithm of retention factor from HPLC_IAM_	HPLC
log P	partition coefficient	Hyperchem
log U/D	describes the extent of ionisation, calculated from pKa, according to the equations: pK_a_—pH (acids) or pH—pK_a_ (bases)	ACD Labs
M/P	milk-to-plasma ratio; ratio between API concentration in milk and plasma	[21,22,23]
M/P_code_	M/P converted to categorical values (code: 1–4)	
NP; NP/C	the R_f_ from NP BSA-modified plate; R_f_ from modified plate/R_f_ from clear one (control)	TLC
PB	protein binding	Drugbank
PB_code_	PB converted to categorical values (code: 1–5)	
PhCharge	charge of a compound in physiological environment	Drugbank
PSA	polar surface area of a molecule	Hyperchem
RP; RP/C	the R_f_ from RP-2 BSA-modified plate; R_f_ from modified plate/R_f_ from clear one (control)	TLC
Sa	surface area of a molecule	Hyperchem

**Table 3 cells-12-00421-t003:** Correlation matrix of the most important molecular descriptors for crossing the BBB.

n = 37	CNS+/−	B2 > −0.9	B2	PhCharge	Acid/Base	B2 > −0.52	PB
CNS+/−	1.00						
B2 > −0.9	0.36	1.00					
B2	0.57	0.77	1.00				
PhCharge	0.01	−0.15	−0.18	1.00			
acid/base	0.28	0.10	0.31	0.71	1.00		
B2 > −0.52	0.53	0.78	0.73	0.00	0.23	1.00	
PB	−0.18	−0.41	−0.45	−0.07	−0.15	−0.26	1.00

**Table 4 cells-12-00421-t004:** Correlation matrix in the CNS+ group.

n = 21	B2	LactMed	LLL H	log U/D	M/P	PB	PB_code_	Acid/Base
B2	1.00							
LactMed	0.085	1.00						
LLL H	0.14	0.099	1.00					
log U/D	−0.14	−0.25	−0.28	1.00				
M/P	0.25	0.16	0.060	−0.18	1.00			
PB	−0.31	−0.067	−0.10	−0.061	−0.42	1.00		
PB_code_	−0.17	0.33	0.28	−0.045	0.52	−0.82	1.00	
acid/base	0.54	0.30	0.21	−0.090	0.54	0.032	0.30	1.00

**Table 5 cells-12-00421-t005:** Internal correlations between chromatographic parameters for CNS+/− cases and CNS+ only.

		**CNS+/−, n = 37**		
	NP	RP	logk(HSA)	logk(IAM)
M/P	−0.30	−0.27	−0.041	−0.031
		**CNS+, n = 25**		
	NP	RP	logk(HSA)	logk(IAM)
M/P	−0.33	−0.42	−0.059	0.24

**Table 6 cells-12-00421-t006:** Internal correlations between chromatographic parameters and control for CNS + APIs.

CNS+, n=25				
	NP	NP/C	RP	RP/C
M/P	–0.33	–0.35	–0.42	–0.57

**Table 7 cells-12-00421-t007:** Molecular descriptors used in random forest regression.

No.	Molecular Descriptor	No.	Molecular Descriptor
1.	M/P	5.	HD
2.	PB	6.	log P
3.	PhCharge	7.	chromatographic data: R_f_ or log k
4.	HA		

**Table 8 cells-12-00421-t008:** RF modelling results, models were created 20 times, and each time different decision trees were involved.

NP TLC	RP TLC
R^2^ = 0.8615; Q^2^ = 0.7915	R^2^ = 0.8484; Q^2^ = 0.7850
CV = 55.725	CV = 56.105
RMSE_CV_ = 0.3279	RMSE_CV_ = 0.3302
*p* = 5.087 × 10^−17^	*p* = 3.454 × 10^−16^

## Data Availability

Not applicable.

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
