# Peer review of "Computational Approach to Drug Penetration across the Blood-Brain and Blood-Milk Barrier Using Chromatographic Descriptors"

_cells, 2023, doi:10.3390/cells12030421_

Round 1

Reviewer 1 Report

Out of my field, but clearly demonstrates that a simple TLC (impregnated with BSA) technique may be a useful predictor of drug penetration into breast milk. Statistical analysis utilising a wide range of molecular descriptors (Table 1) is thorough.

Very minor corrections:

Page 1, 2nd paragraph.......'be find' should be 'be found'.

Some issues with referencing.....7 times error message...'Reference source not found'.

Author Response

Thank you for your review. The typo on the first page has been corrected.

As for the bibliography, it was created with the help of Mendeley and I do not see such errors on my part. If necessary, I will send the bibliography to the Editor in a different format. 

Reviewer 2 Report

The concept of this paper is interesting; very little has been published on computational assessments of blood-milk barrier penetration. However, there are several issues with thae manuscript which the authors need to address.

# General: Add a few sentences describing similarities and differences between the BBB and blood/milk barrier. The immobilized artificial membrane column chromatography miust also be explained.

# p.2, introduction: "Determining this parameter [i.e., M/P] in vivo is even more difficult than log BB (mainly for ethical reasons),..." Why? Breast milk is easy to obtain non-invasively, and if the woman is not breastfeeding she can participate in such simple studies.

# The identity of the 37 considered APIs is not disclosed.

# Why was BSA, and not HSA - used to impregnate the TLC plates?

# How was drug binding to the albumins corrected for? The affinity of APIs to plasma proteins is a central point of the argumentation line.

# Citation system errors ("Error! Reference source not found.") are present throughout, which makes it difficult to assess the appropriateness of the provided references.

Author Response

Thank you for your review. I applied all the suggested changes (marked in red in the manuscript):

  1. Add a few sentences describing similarities and differences between the BBB and blood/milk barrier. The immobilized artificial membrane column chromatography miust also be explained.

Ad 1. The differences between both barriers have been explained on the 2nd page, third paragraph (lines 58-61). IAM chromatography is shortly explained in the 6th paragraph, 2nd page as well.

  1. 2, introduction: "Determining this parameter [i.e., M/P] in vivo is even more difficult than log BB (mainly for ethical reasons),..." Why? Breast milk is easy to obtain non-invasively, and if the woman is not breastfeeding she can participate in such simple studies.

Ad 2. Page 2, second paragraph, the part with the explanation has been added: “Difficulties in obtaining the M/P ratio in vivo are mainly due to ethical reasons and the danger of affecting the breastfed infant, therefore most approaches to the subject are case studies. Only a few results are obtained for individual drugs and may these not reflect what is happening in the scale of the entire population. The problem is even greater with drugs newly introduced to the market [10]. The computational methods could be helpful in estimation of M/P ratios based on API properties and general principles of permeation through biological membranes.”

  1. The identity of the 37 considered APIs is not disclosed.

Ad 3. 37 APIs are listed in the Supplementary material, along with their properties used in statistical modelling. Additionally, their structures have been enclosed as well.

  1. Why was BSA, and not HSA - used to impregnate the TLC plates?

Ad 4. Bovine serum albumin reveals similar properties the human version, but is much cheaper, therefore BSA is used in many experiments. Explanation, with an adequate citation has been added to the manuscript (pg 2, 5th paragraph)

  1. How was drug binding to the albumins corrected for? The affinity of APIs to plasma proteins is a central point of the argumentation line.

Ad 5. I'm afraid I didn't fully understand the question. The albumin is the most abundant protein in the bloodstream and is responsible for most of the plasma protein binding. A necessary simplification in the construction of a simple QSPR model was the assumption that PB equals binding to albumin. I hope that was the subject of that question.

  1. Citation system errors ("Error! Reference source not found.") are present throughout, which makes it difficult to assess the appropriateness of the provided references.

Ad 6. The bibliography was established by Mendeley. I do not see these errors in my version of the manuscript, but I understand that they may have occurred. The list of references has been cut and pasted as plain text, with no formatting options, and I hope this solves the problem.